# Mangiferin Ameliorates Obesity-Associated Inflammation and Autophagy in High-Fat-Diet-Fed Mice: In Silico and In Vivo Approaches

**DOI:** 10.3390/ijms232315329

**Published:** 2022-12-05

**Authors:** Ji-Won Noh, Han-Young Lee, Byung-Cheol Lee

**Affiliations:** Department of Clinical Korean Medicine, Graduate School, Kyung Hee University, 26 Kyungheedae-to, Dongdaemun-gu, Seoul 02447, Republic of Korea

**Keywords:** *Anemarrhenae rhizoma*, mangiferin, inflammation, autophagy, insulin resistance, obesity

## Abstract

Obesity-induced insulin resistance is the fundamental cause of metabolic syndrome. Accordingly, we evaluated the effect of mangiferin (MGF) on obesity and glucose metabolism focusing on inflammatory response and autophagy. First, an in silico study was conducted to analyze the mechanism of MGF in insulin resistance. Second, an in vivo experiment was conducted by administering MGF to C57BL/6 mice with high-fat-diet (HFD)-induced metabolic disorders. The in silico analysis revealed that MGF showed a high binding affinity with macrophage-related inflammatory cytokines and autophagy proteins. In the in vivo study, mice were divided into three groups: normal chow, HFD, and HFD + MGF 150 mg/kg. MGF administration to obese mice significantly improved the body weight, insulin-sensitive organs weights, glucose and lipid metabolism, fat accumulation in the liver, and adipocyte size compared to HFD alone. MGF significantly reduced the macrophages in adipose tissue and Kupffer cells, inhibited the gene expression ratio of tumor necrosis factor-α and F4/80 in adipose tissue, reduced the necrosis factor kappa B gene, and elevated autophagy-related gene 7 and fibroblast growth factor 21 gene expressions in the liver. Thus, MGF exerted a therapeutic effect on metabolic diseases by improving glucose and lipid metabolism through inhibition of the macrophage-mediated inflammatory responses and activation of autophagy.

## 1. Introduction

The prevalence of metabolic diseases, such as obesity, arteriosclerosis, hyperlipidemia, and diabetes, is increasing worldwide [1]. Insulin resistance is a common pathogenic event in type 2 diabetes, obesity, non-alcoholic fatty liver disease, and cardiovascular disease. Fat accumulation in the liver and muscles, where high proportions of glucose and lipid metabolism occur, is the main reason for obesity-induced insulin resistance [2,3]. Additionally, obesity-induced chronic inflammation is strongly associated with the pathogenesis of insulin resistance [4]. An increased adipose tissue accumulation due to excessive energy consumption can cause inflammatory responses and insulin signaling disorders by increasing the secretion of inflammatory cytokines from macrophages [5,6]. Recently, studies on the relationship of autophagy with metabolic diseases have been increasing [7]. Suppression of autophagy-related gene 7 (ATG7) impairs insulin signaling, whereas restoration of ATG7 expression improves insulin resistance [8].

The rhizome of *Anemarrhenae asphodeloides* belongs to the family Liliaceae and is cultivated in Korea, Mongolia, China, and other Eastern countries. It is 0.5–1.5-cm wide and disc-shaped and diverges its surroundings. Anemarrhenae Rhizoma has been used to treat febrile diseases accompanied by high fever, thirst, dry cough, fever, and diabetes [9]. A review focused on pharmacological and biological effects of MGF on metabolic disorders showed that MGF, an Anemarrhenae Rhizoma polyphenol, alleviates obesity and fatty liver, protects pancreatic β-cells, and improves diabetes complications by controlling the crosstalk between lipid and carbohydrate metabolisms [10,11]. MGF exerts anti-inflammatory effects by inhibiting interleukin-1 receptor-associated kinase 1 (IRAK1) phosphorylation in the mitogen-activated protein kinase (MAPK) and nuclear factor-κB (NF-κB) pathways in C57BL/6 mice with chemically induced inflammatory bowel disease (IBD) with MGF administered at a dose of 50 mg/kg body weight for 10 days. Additionally, it exerts liver protective effects by scavenging reactive oxygen species and regulating the expression of inflammatory cytokines in galactosamine-exposed hepatotoxic Swiss albino male rats with six doses of MGF (5, 10, 15, 20, and 25 mg/kg body weight/day for 14 days) [12,13]. MGF suppresses the mTOR activity in the liver and hepatic de novo lipogenesis in male Kunming mice with different doses of MGF (15, 30, and 60 mg/kg body weight/day for 12 weeks) [14]. However, few studies investigated the role of MGF treating obesity in aspect of inflammation and autophagy.

To understand the effect of MGF on insulin resistance and its possible mechanism in a different manner, we first designed in silico experiments, including screening of target genes, functional enrichment analysis, and molecular docking. Subsequent in vivo experiments were conducted to precisely identify the effects and mechanisms of MGF on obesity-associated inflammation and autophagy by evaluating the metabolic phenotypes, the population of macrophages, and gene expressions in the liver and fat.

## 2. Results

### 2.1. In Silico Network Pharmacology Approaches of MGF

#### 2.1.1. Hub Genes of MGF and Insulin Resistance

Seventy-two target genes and 8628 insulin resistance target genes were identified. Fifty-nine overlapping genes were identified between the target genes of MGF and insulin resistance (Figure 1a). 

#### 2.1.2. Analysis of Functional Enrichment

Gene ontology (GO) enrichment analysis with 59 overlapping genes was performed, including 43 molecular function (MF), 154 biological process (BP), and 27 cellular component (CC) pathways. The top 10 enriched BP, top 5 enriched MF, and top 5 CC terms were extracted. Twenty GO terms were visualized using a dot plot (Figure 1b). In the BP category, the following were selected: inflammatory response, signal transduction, oxidation–reduction process, peptidyl-serine phosphorylation, negative regulation of apoptosis, protein phosphorylation, positive regulation of the ERK1 and ERK2 cascade, apoptosis, positive regulation of NF-κB transcription factor activity, and positive regulation of I-κB kinase/NF-κB signaling. In the MF category, the following functions were selected: zinc ion binding, ATP binding, enzyme binding, protein kinase activity, and protein kinase binding. In the CC category, the following components were listed: plasma membrane, an integral component of the membrane, cytoplasm, cytosol, and extracellular exosomes. Kyoto Encyclopedia of Genes and Genomes (KEGG) pathway enrichment analysis was performed with 59 overlapped genes, and 17 insulin-related pathways were selected, including inflammatory mediator regulation of transient receptor potential (TRP) channels, sphingolipid signaling, calcium signaling, Ras signaling, hypoxia-inducible factor-1 signaling, thyroid hormone signaling, cAMP signaling, Rap1 signaling, MAPK signaling, and PI3K-AKT signaling pathway, aldosterone synthesis and secretion, insulin resistance, insulin signaling pathway, type 2 diabetes mellitus, insulin secretion, NF-κB signaling, and mTOR signaling pathway (Figure 1c). These results imply that MGF is associated with multiple pathways related to insulin resistance.

#### 2.1.3. Docking Simulation

Molecular docking of MGF with tumor necrosis factor-α (TNF-α), interleukin-6 (IL-6), nuclear factor kappa B (NF-κB), AKT1, ATG7, and fibroblast growth factor 21 (FGF21) showed high affinity (Figure 2a–f). The highest binding energies were −9.2, −6.6, −6.8, −7.4, −7.8, and −6.5 kcal/mol, respectively.

### 2.2. In Vivo Experiments of the Effects of MGF on Glucose and Lipid Metabolism

#### 2.2.1. Effects of MGF on Weight Gain 

First, food intake and body weight (BW) of each group were measured to assess the effects of MGF. A significant difference was found in comparing liver, body, and epididymal fat weights between the high-fat diet (HFD) and normal chow (NC) groups at week 14, as the former group was higher than the latter. (BW, 51.92 ± 1.32 g vs. 28.86 ± 1.29 g, *p* < 0.001; epididymal fat, 2.30 ± 0.18 g vs. 0.41 ± 0.04 g, *p* < 0.001; liver, 2.26 ± 0.18 g vs. 1.18 ± 0.05 g, *p* < 0.001). Compared to the HFD group, the MGF group had significantly lower BW gain, epididymal fat, and liver weight (BW change, 25.20 ± 1.04 vs. 31.62 ± 1.03 g, *p* < 0.01; epididymal fat, 1.74 ± 0.12 vs. 2.30 ± 0.18 g, *p* < 0.05; liver, 1.19 ± 0.14 vs. 2.26 ± 0.18 g, *p* < 0.01; Figure 3a–d). The HFD group showed significantly more calories per day than the NC group (24.19 ± 3.76 g vs. 9.21 ± 0.64 g, *p* < 0.001). The calories consumed by the MGF group did not differ significantly from the HFD group (Figure 3). 

#### 2.2.2. Effects of MGF on Glucose Metabolism

To evaluate the effect of MGF on glucose metabolism, fasting blood glucose (FBG) and blood glucose levels at each timepoints were assessed with the oral glucose tolerance test (OGTT). The FBG level was significantly higher in both HFD and MGF groups than in the NC group (HFD, 225 ± 18.77 mg/dL vs. 106 ± 5.65 mg/dL, *p* < 0.001; MGF, 166 ± 12.24 mg/dL vs. 106 ± 5.65 mg/dL, *p* < 0.01). The MGF group showed a significantly lower FBG level than the HFD group (166 ± 12.24 mg/dL vs. 225.00 ± 18.77 mg/dL, *p* < 0.05; Figure 4a). In the case of the OGTT, blood glucose levels reached a peak at 30 min in all groups. The blood glucose levels at 30 and 60 min were significantly lower in the MGF group compared to the HFD group (30 min, 284 ± 14.70 mg/dL vs. 404 ± 12.96 mg/dL, *p* < 0.001; 60 min, 209 ± 18.87 mg/dL vs. 271 ± 12.14 mg/dL, *p* < 0.05; Figure 4b). The area under the curve (AUC) for the HFD group was significantly higher than that for the NC or MGF group (Figure 4c). The MGF group showed significantly lower incremental area under the curve (iAUC) than the NC and HFD group (Figure 4d).

The fasting insulin level was significantly elevated in the HFD group compared to the NC (3.74 ± 0.59 ng/dL vs. 1.17 ± 0.26 ng/dL, *p* < 0.01) and MGF (2.04 ± 0.35 ng/dL vs. 3.74 ± 0.59 ng/dL, *p* < 0.05) groups. The HFD group had significantly higher HOMA-IR compared to the NC (62.76 ± 15.00 vs. 8.92 ± 1.99, *p* < 0.01) and MGF (24.99 ± 5.43 vs. 62.76 ± 15.00, *p* < 0.05) groups. HOMA-IR increased significantly in the MGF group than in the NC group (24.99 ± 5.43 vs. 8.92 ± 1.99, *p* < 0.05), but the fasting insulin level did not differ significantly (Figure 4e,f).

#### 2.2.3. Effects of MGF on Lipid Metabolism

Total cholesterol (TC), low-density lipoprotein (LDL) cholesterol, nonesterified fatty acids (NEFA), and phospholipid levels were significantly higher in the HFD group than in the NC group (TC, 203.6 ± 12.68 mg/dL vs. 110.2 ± 3.17 mg/dL, *p* < 0.001; LDL cholesterol, 33.6 ± 3.54 mg/dL vs. 11.8 ± 0.80 mg/dL, *p* < 0.001; NEFA, 2762.8 ± 32.61 mEq/L vs. 2361.0 ± 55.55 mEq/L, *p* < 0.001; phospholipid, 310.2 ± 13.11 mg/dL vs. 244.8 ± 1.88 mg/dL, *p* < 0.01; Figure 5a–e). The NEFA level was significantly lower, while the high-density lipoprotein (HDL) cholesterol level was significantly higher in the MGF group than in the HFD group (HDL cholesterol, 153.62 ± 3.14 mg/dL vs. 108.80 ± 9.39 mg/dL, *p* < 0.01).

In the oral fat tolerance test (OFTT), the triglyceride (TG) level of each group peaked at 120 min and then steadily declined. A significant difference was found in comparing the TG level between HFD and NC groups at every measurement time except 360 min, as the former group was higher than the latter (Figure 5f). The MGF group showed a lower TG level at every measurement time than the HFD group, and significantly lower at 120 min (303.2 ± 28.00 mg/dL vs. 395.0 ± 17.59 mg/dL, *p* < 0.05) and 240 min (191.4 ± 15.79 mg/dL vs. 287.8 ± 20.11 mg/dL, *p* < 0.01). The MGF group showed a higher TG level at 120 min than the NC group but without statistical significance (Figure 5f). When three groups were compared for TG AUC, a significantly high outcome was found in the HFD group than in the MGF group (76,440 ± 4193.22 mg∙min/dL vs. 101,904 ± 4,454.11 mg∙min/dL, *p* < 0.01; Figure 5g).

#### 2.2.4. Effects of MGF on Adipose Tissue

The total proportion of CD45 (+) and CD68 (+) adipose tissue macrophages (ATMs) were significantly lower in the MGF group than in the HFD group (64.47 ± 1.90 vs. 75.69 ± 0.60, *p* < 0.01; Figure 6a,b). In the subgroup analysis of ATMs, the HFD group presented a significantly increased CD11c (+) M1 ATM subtype and decreased CD206 (+) M2 ATM subtype. However, the MGF group showed significantly decreased M1 ATMs compared to the HFD group (59.67 ± 2.44 vs. 47.26 ± 2.27, *p* < 0.01; Figure 6c–e). 

F4/80 gene expression in adipose tissue was significantly lowered in both NC and MGF groups (NC, 0.99 ± 0.03 vs. 10.68 ± 1.18, *p* < 0.001; MGF 6.59 ± 0.86 vs. 10.68 ± 1.18, *p* < 0.05) compared to that in the HFD group (Figure 6f). A significant decrease in TNF-α was found in the MGF group compared to its level in the HFD group (2.01 ± 0.23 vs. 9.20 ± 2.36, *p* < 0.05; Figure 6g). The IL-6 expression was significantly lowered in the NC group (HFD, 1.00 ± 0.00 vs. 2.03 ± 0.39, *p* < 0.05; MGF, 1.00 ± 0.00 vs. 1.92 ± 0.31, *p* < 0.05) than both HFD and MGF groups, but no significant difference was found between the HFD and MGF groups (Figure 6h).

#### 2.2.5. Effects of MGF on Kupffer Cells (KCs) and Gene Expressions

The total percentage of KCs was significantly higher in the HFD group compared to the NC group, while the MGF group showed a significantly decreased KC proportion compared to the HFD group (31.76 ± 1.47 vs. 44.61 ± 2.26, *p* < 0.001; Figure 7a,b). 

The NF-κB gene expression level in the liver was significantly higher in the MGF group (2.73 ± 0.37 vs. 5.20 ± 0.58, *p* < 0.05) than in the HFD group (Figure 7c). A significant difference was found in a comparison of AKT1 gene expression levels only between HFD and NC groups (Figure 7d). When the three groups were compared for ATG7 gene expression level, a significantly low outcome was discovered in the HFD group than in the NC group and MGF group (NC, 0.61 ± 0.10 vs. 1.01 ± 0.03; MGF, 0.61 ± 0.10 vs. 1.00 ± 0.13, *p* < 0.05; Figure 7e). FGF21 expression was significantly higher in the MGF group than in the HFD group (6.04 ± 0.73 vs. 2.92 ± 0.71, *p* < 0.05; Figure 7f).

#### 2.2.6. Effects of MGF on Adipocyte and Hepatic Fat Area

When the three groups were compared for the size of adipocytes and fat area in the liver, a significantly high outcome was discovered in HFD group than of the NC group (fat: 16,122 ± 1577.0 vs. 4659 ± 225.4, *p* < 0.001; liver: 65.49 ± 4.12 vs. 13.54 ± 0.05, *p* < 0.001) and the MGF group (fat: 16,122 ± 1577.0 vs. 9438 ± 527.1, *p* < 0.001; liver: 65.49 ± 4.12 vs. 21.88 ± 0.08, *p* < 0.001; Figure 8a–c).

## 3. Discussion

This study aimed to investigate the underlying mechanisms of MGF in insulin resistance at molecular and genetic level focusing on the inflammatory response and autophagy, and new findings were obtained through in silico and in vivo experiments on MGF. First, the relationship between MGF and NF-κB signaling, MAPK signaling pathway, and inflammatory response explored by in silico experiments signified the possibility of improving insulin resistance by regulation of inflammation. Taking the results from the in silico, in vivo experiments specifically studied BW, ATMs, KCs, and mRNA expression of F4/80, TNF-α, IL-6, and NF-κB. Second, in silico experiments investigating the association between MGF and insulin signaling and the PI3K-AKT signaling pathways revealed the possibility of improving insulin resistance through a decrease in fat accumulation. The in vivo experiment demonstrated that MGF decreased the weight and fat areas of the fat pad and liver and increased the mRNA expressions of FGF21 compared to the HFD group. Finally, in silico experiments, it was revealed that MGF affected autophagy-related mechanisms such as mTOR signaling pathways among the genes related to insulin resistance. Consistent with the in silico results, MGF modulated the mRNA expressions of ATG7 and FGF21 in the in vivo experiments. Our study suggested that the action mechanism of MGF on obesity-associated abnormal glucose and lipid metabolism might be inhibiting inflammation in adipose tissue, enhancing autophagy, and manipulating ATMs and KCs. Therefore, the potential of MGF as a therapeutic agent for metabolic disorders is highlighted through the results. 

According to previous studies, the anti-inflammatory and anti-diabetic effects of MGF by inhibiting IRAK1 phosphorylation in NF-κB and MAPK pathways [12], tumor growth and inflammatory responses-related pathways [14], inhibition of JNK activation that can cause liver lipid deposition and insulin resistance [15], and renal fibrosis inhibition via PTEN/PI3K/AKT pathway have been reported [16]. Therefore, this study explored the MGF effect on the possible causes of insulin resistance, including increased fat mass and hepatic lipid accumulation, inflammatory responses mediated by macrophages, and dysregulation of autophagy.

In silico experiments revealed hub pathways of overlapped genes between insulin resistance and MGF, and showed that the significant pathways were NF-κB, MAPK, and PI3K-AKT signaling [17]. Because of few previous studies about these signaling pathways, the interactions between MGF and ATG7 and FGF21 were identified by molecular docking simulation. Sequential in vivo experiments described the effects of MGF on weight, glucose and lipid metabolism, inflammatory response in adipose tissues, and autophagy in the liver of HFD-fed mice.

Molecular docking techniques indicated MGF-protein interactions, which were evaluated based on binding affinity. As TNF-α and NF-κB were among the top 10 hub genes, and the functional analysis of MGF was related to inflammation in macrophages and autophagy, several genes were selected for docking simulation. NF-κB, TNF-α, IL-6, and macrophage marker F4/80 are inflammation-related substances [18], and NF-κB is essential for an inflammatory response in the hepatic tissue [19]. The reduced activities of PI3K and AKT, which affects insulin resistance results from increased usage of free fatty acids in tissues, reduced glucose inflow from muscles, and lack of inhibition of hepatic glucose output [5]. Molecular interactions showed a high binding affinity of MGF with TNF-α, IL-6, NF-κB, AKT, ATG7, and FGF21. A high binding relationship between MGF and their target proteins demonstrates therapeutic efficacy [20]. Therefore, these molecular docking results suggest that the mechanisms of MGF in insulin resistance might be related to inflammation and autophagy.

In the current study, significant reduction in weight gain was observed in the MGF group, indicating the effectiveness of MGF in preventing obesity. The epididymal fat pads in mice are a major indicator for evaluating changes in white adipose tissue (WAT) because of the secretion of various adipokines and insulin resistance [21,22]. Liver weight also is an indirect indicator of obesity and inflammatory conditions, as fat accumulation in obesity due to high-fat diet accounts for liver weight [23]. The epididymal fat and liver weights were significantly lower in the MGF group than in the HFD group. The obesity preventive effects were not based on differences in food intake but on significant immune-modulating benefits, including decreased M1 ATM, and inhibition of TNF-α and NF-κB. The WAT increase is closely linked to obesity, and fat accumulation in the liver is affected by the increased NEFA from WAT lipolysis connected through the hepatic portal vein [6]. In obesity, lipids from high lipolysis of enlarged fat are accumulated in KCs before the number of KCs increased. Fat-laden KCs promote TG accumulation in liver and develop the severity of NAFLD by secreting pro-inflammatory cytokines, including TNF-α and IL-1β which suppress PPARα pathway via NF-κB activation [24]. We observed that MGF group showed significantly low fat weight, decreased adipocyte size and less fat accumulation in the liver along with the reduced number of KCs and down-regulation of NF-κB. 

When hyperglycemia is maintained, a non-enzymatic reaction occurs between blood glucose and proteins or lipoproteins, resulting in advanced glycation end-products (AGEs). AGEs intensify the pathological aging of blood vessels; thus, controlling blood glucose is essential for people with diabetes and obesity [25,26]. The FBG levels were significantly decreased in the MGF group. The OGTT results showed significantly lower glucose levels in the MGF group than in the HFD group when the levels were measured at 30 and 60 min. In addition, fasting serum insulin levels and HOMA-IR in the MGF group were significantly lower than in the HFD group, indicating the effectiveness of MGF in improving glucose metabolism. 

Although HFD leads to high TG levels, the enlarged adipose tissue induces high lipolysis, leading to a high level of circulating NEFA, which comes into the liver, resulting in high hepatic TG levels [27]. In diabetes, the serum lipoprotein concentration also increases due to the disorder of insulin-inhibiting lipoprotein production [28]. Obesity-induced dyslipidemia is characterized by elevated NEFA, TG, and LDL-C levels and reduced HDL-C levels [29]. In this study, the serum lipids of the HFD group consisted of elevated NEFA, TG, TC, LDL-C, and HDL-C levels, same as the typical lipid profile in obesity. However, the MGF administration significantly reduced NEFA and TG levels and elevated HDL-C levels. Moreover, the NEFA and TG levels in the MGF group were not statistically different from those in the NC group. In OFTT, TG levels in the MGF group were significantly lower at 120 and 240 min compared to those in the HFD group. These results demonstrate that MGF can improve obesity-induced dyslipidemia.

Obesity, a low-grade, chronic inflammatory condition, increases ATMs and abnormal cytokine secretion and impairs macrophage metabolism by increasing the M1 phenotype, which acts as an obstacle to insulin signaling [30]. F4/80 is a marker that allows the screening of mononuclear phagocytes and is known to be strongly correlated to body mass index and adipocyte size [6]. When the number of pro-inflammatory M1 ATMs is increased in obesity, TNF-α and IL-6 are highly secreted, which are involved in the insulin resistance of adipocytes by inhibiting insulin signaling and stimulating lipolysis of adipocytes. In the MGF group, total ATM, M1 ATM, F4/80, and TNF-α levels were significantly reduced compared to those in the HFD group. 

Obesity-induced insulin resistance results in abnormal elevated circulating insulin, promotion of fatty acid synthesis, and increase in FGF21 ultimately causing a FGF21-resistant state. Liver-derived FGF21 signaling in adipose tissue is essential for maintaining insulin sensitivity and lipolysis. Higher liver-derived FGF21 levels can lower BW, blood glucose and liver fat [31], and activate ATG7 expression which can improve the impairment of autophagy and hepatosteatosis [32]. ATG7, a vital autophagic element, contributes to insulin signaling and regulates hepatic insulin sensitivity by ameliorating systemic glucose homeostasis and endoplasmic reticulum stress [8]. Recently, it was suggested that the possibility of increased autophagy activity induced by obesity might act as a compensatory signal to reduce insulin resistance and inflammatory responses in adipose tissue [33]. The mRNA expression of NF-κB, AKT, ATG7, and FGF21 in the liver tissue was analyzed to confirm the effects of MGF on the linkage of autophagy and insulin resistance, and the current results explained that MGF decreased inflammation and regulated autophagy. In the MGF group, the mRNA expression of ATG7 and FGF21 was significantly increased compared to that in the HFD group, and NF-κB was significantly decreased. NF-κB is vital for inflammatory responses and stress signals. However, AKT expression increased with no significance in the MGF group compared to the HFD group. AKT plays a role in multiple pathways by inhibiting glucose production [34]. Reduced activation of PI3K and AKT decrease glucose inflow mediated by GLUT4 in muscles, and cause insulin resistance due to dysregulation of hepatic glucose output in liver [5]. However, unlike prior studies, our data revealed no significant difference in AKT between the MGF and HFD groups [35]. Controversial results between the current analysis and previous studies emphasize the need for additional research. 

## 4. Materials and Methods

### 4.1. In Silico Research of the Effects of MGF on Insulin Resistance

#### 4.1.1. Search for MGF Target Genes

We searched the SMILES database of MGF from PubChem (pubchem.ncbi.nlm.nih.gov/ (accessed on 18 March 2021)). MGF target genes were collected using SwissTargetPrediction (swisstargetprediction.ch) based on MGF’s SMILES, and data with a zero probability were eliminated.

#### 4.1.2. Search for Overlapping Target Genes between MGF and Insulin Resistance

We collected the target genes of insulin resistance from GeneCards (genecards.org/ (accessed on 18 March 2021)) and used the term “insulin resistance”. Relevance scores > 10 from GeneCards were selected. Overlapping genes between insulin resistance and MGF were obtained, and VENNY 2.1 (bioinfogp.cnb.csic.es/tools/venny/ (accessed on 18 March 2021)) was used to represent the overlapping genes in a Venn diagram.

#### 4.1.3. Analysis of Functional Enrichment

To obtain GO and KEGG data, overlapping genes were analyzed using DAVID (david.ncifcrf.gov/ (accessed on 18 March 2021)). Each CC, BP, and MF from GO terms were sorted by gene count, and the top 10 BP, top 5 CC, and MF were selected. Gene count sorted the KEGG pathways, and 17 pathways related to insulin resistance were determined. The role of the obtained data was analyzed using R software ver. 4.2. The results were visualized with dot plots indicating the enrichment terms.

#### 4.1.4. Docking Simulation

The binding site of MGF to target proteins and its binding affinity were investigated to evaluate the interactions between target proteins and MGF. As the ligand, we chose MGF, TNF-α, and NF-κB, which are highly related to vital pathways of the functional enrichment analysis. IL-6, AKT1, and FGF21 were also selected as receptors to evaluate their effects on insulin resistance, and ATG7 was chosen as a receptor to assess its effects on autophagy. We obtained 3D data for MGF (CID 5281647) from PubChem (pubchem.ncbi.nlm.nih.gov/ (accessed on 18 March 2021)). The structures of TNF-α (2AZ5), IL-6 (1ALU), NF-κB (1NFI), AKT1 (1UNP), ATG7 (3T7H), and FGF21 (6M6E) were obtained from the PDB (rcsb.org). Target proteins were pretreated using the Biovia Discovery Studio Visualizer, deleting unnecessary domains and adding polar groups. Molecular docking and binding affinity analysis of MGF to TNF-α, IL-6, NF-κB, AKT1, ATG7, and FGF21 were performed using Pyrx and Autodock VINA. Visualization of the binding structures was performed using the Biovia Discovery Studio Visualizer.

### 4.2. In Vivo Experiments to Explore the Effects of MGF on Insulin Resistance 

#### 4.2.1. Animals and Drug

Experimental C57BL/6 male mice, weighing 19–21 g, 6 weeks old, were obtained (Central Lab Animals Inc., Seoul, Republic of Korea). With 40 to 70% humidity and controlled lighting (12-h light/dark cycle), water and regular rodent chow were supplied for the free approach. After 1 week of adaptation, HFD and MGF groups received HDF containing 60% fat of total calories (D12492 (Appendix A), Research Diets, New Brunswick, NJ, USA). MGF (CAS RN:4773-96-0) was purchased from Sigma-Aldrich^®^ Chemicals (https://www.sigmaaldrich.com/KR/ko (accessed on 10 April 2021)).

#### 4.2.2. Study Design and Drug Administration

A total of fifteen mice were simultaneously randomized to the three groups without considering any other variable: NC, HFD (control), and MGF 150 mg/kg groups. HFD and MGF groups were administered with HFD for 14 weeks to induce obesity. After 6 weeks of HFD, none of the groups showed a significant difference except for the BW in the NC group. Subsequently, MGF 150 mg/kg/day was orally administered to the MGF group, and normal saline was administered to the HFD and NC groups using a gavage needle for 8 weeks.

#### 4.2.3. Measuring BW, Oral Intake, and Liver and Fat Tissues

We measured BW weekly using an electronic balance (CAS 2.5D; Seoul, Republic of Korea) in the morning at the same time. Each mouse was placed on a plastic bowl during BW recording session, and to minimize the error during the measurement, BW was recorded once the mice was stable. The quantity consumed by each group was computed every morning by the difference between the day before feed and the following day using an electronic balance (CAS 2.5D; Seoul, Republic of Korea). At the end of the study, liver and epididymal fat pads were weighed using an electronic balance.

#### 4.2.4. OGTT and Insulin Resistance Measurement

OGTT was performed at week 11. After fasting for 14 h, the mice were orally fed 2 g/kg of glucose liquefied in distilled water. The blood glucose level was evaluated at 0, 30, 60, 90, 120, and 180 min using a strip-operated blood glucose sensor (AccuCheck Performa, Basel, Switzerland) with blood samples from tail vein.

After 14 weeks, blood was collected from 6-h fasting mouse’s tail vein to measure blood glucose levels and insulin concentration. BD Microtainer was used to collect a blood sample at room temperature, followed by 20 min of centrifuging at 2000 G to isolate the serum. An ultrasensitive insulin ELISA kit (Crystal Chem Inc., Elk Grove Village, IL, USA) was used to measure the serum insulin levels. Insulin standards and samples were dispensed 5 μL each into antibody-coated microplates. After 2 h of incubation (at 4 °C), we washed the samples five times and conjugated them with the anti-insulin enzyme reacted with the enzyme substrate solution. After 10 min, the reaction stop solution was added, and the insulin level was measured at 450 nm. 

#### 4.2.5. OFTT and Lipid Analysis

At week 12 of the trial, OFTT was performed after 14 h of fasting. After the mice were administered with 2 mL/kg of olive oil, TG levels were assessed by collecting tail vein blood at 0, 120, 240, and 360 min. Accutrend Plus (Roche, Basel, Switzerland) was used to evaluate the TG concentrations. At week 14, we evaluated TC, phospholipids, NEFA, LDL cholesterol, and HDL cholesterol with blood samples from the heart.

#### 4.2.6. Analysis of Gene Expression in Adipose Tissue

The mice were euthanized at week 14, and epididymal fat pads were dissected and wrapped with aluminum foil. Then, samples were stored in liquid nitrogen at −70 °C until RNA extraction. Mini RNA Isolation IITM (ZYMO RESEARCH, Irvine, CA, USA) was used to isolate RNA from adipose tissue. Defrosted adipose tissue was transferred to tubes with 300 µL aliquots of the ZR RNA buffer and pulverized by a homogenizer. After centrifugation at 1000 rpm, the supernatant was moved into a column, which was mixed with 350 µL of the RNA wash buffer. Fifty microliters of RNA-free water were added for centrifugation, and the extracted RNA was stored at −70 °C.

#### 4.2.7. Analysis of Gene Expression in Liver Tissue

To examine the expressions of NF-κB p65, AKT, ATG7, and FGF21, quantitative real-time polymerase chain reaction (qRT-PCR) was used. Before qRT-PCR, we prepared complementary DNA (cDNA) using the Advantage RT for PCR Kit (Clontech, Mountain View, CA, USA). Oligo (dT), RNase-free H_2_O, and 1 μg of RNA extracted from the tissue were added and incubated at 70 °C for 2 min. Then, MMLV reverse transcriptase, 10 nM dNTP, recombinant RNase inhibitor, and 5× reaction buffer were added and incubated at 42 °C for 60 min and at 94 °C for 5 min. Reverse transcription PCR was mixed with primers, 2× SYBR Reaction buffer, and d H_2_O to obtain cDNA. The 7900 HT Fast Real-Time PCR System (Applied Biosystems^®^, Waltham, MA, USA) was used for qRT-PCR. The primers used were as follows; in case of F4/80 5′-CTTTGGCTATGGGCTTCCAGTC-3′; 5′-GCAAGGAGGACAGAGTTTATCGTG-3′, in case of TNF-α 5′-TTCTG TCTAC TGAAC TTCGG GGTGA TCGGT CC-3′, 5′-GTATG AGATA GCAAA TCGGC TGACG GTGTG GG-3′, in case of IL-6 5′-AACGATGATGCACTTGCAGA-3′, 5′-GAGCATTGGAAATTGGGGTA-3′, in case of NF-κB p65 5′-ACCACTGCTCAGGTCCACTGTC-3′, 5′-GCTGTCACTATCCCGGAGTTCA-3′, in case of AKT 5′-GAAGACCCAAAGACCAAGATGC-3′, 5′-TCTGACAACAAAGCAGGAGGTG-3′, in case of AKT 5′-CAGTTTGCCCCTTTTAGTAGTGC-3′, 5′-CTTAATGTCCTTGGGAGCTTCA-3′, in case of FGF21 5′-AAACGCGTTCTGGGAACCTCACAGCTCA-3′ and 5′-CCAGATCTCAGGGCTGCGCTCCGTTC-3′ were used. Each gene of cycle threshold (Ct) value was obtained using the SDS Software 2.4 (Applied Biosystems^®^, Waltham, MA, USA), and altered to relative quantitation based on EF-1 α, and fold-change was evaluated. The fold-change value was converted according to the NC group, which was regarded as 1.

#### 4.2.8. Histological Analysis of the Epididymal Fat Pad and Liver

We fixed the fat and liver samples in 10% neutral-buffered formalin and immersed them in 70, 80, 90, and 100% ethanol. The samples were then embedded in paraffin, sliced into 4-μm sections using a microtome, and arranged on gelatin-coated slides. To stain tissues, each slide was dewaxed in xylene, and then 70, 80, 95, and 100% ethanol and distilled water were used to rehydrate the tissues. Hematoxylin and eosin were used to stain the rehydrated tissue, and the histological image of the stained sample was visualized under a high-resolution camera-mounted optical microscope (Olympus BX-50, Olympus Optical, Tokyo, Japan). The fat areas of epididymal fat tissue and liver tissue were measured using ImageJ software.

#### 4.2.9. Isolation of Stromal Vascular Cells (SVCs) and Liver Immune Cells

The dissected epididymal fat pads were soaked in PBS/2% BSA solution, cut out into 1–2-mm size, and mixed with collagenase (Sigma, St. Louis, MO, USA) and DNase I (Roche, Basel, Switzerland). After shaking and removing undigested tissue, PBS and 2% FBS were added to pellet, and the sample was filtered by a 100-μm filter to obtain the SVCs. 

The liver sample without the gall bladder was prepared soaking in the RPMI 1640 medium with 100 mL/L fetal calf serum and squashed through a 200-G mesh filter. Subsequently, Percoll, PBS, and heparin were added before centrifugation. The supernatant was incubated in 1X ACK lysis buffer (Lonza) to dissolve red blood cells, and unnecessary tissue was discarded after centrifugation. 

#### 4.2.10. Fluorescence-Activated Cell Sorting (FACS) of Macrophages

The EDTA sample with 10^5^ of cells by Cellometer (Nexcelom Bioscience LLC, Lawrence, MA, USA) was mixed with FcBlock (BD Pharmingen, Franklin Lakes, NJ, USA) and reacted with fluorophore-conjugated antibodies for 20 min with light limitation. ATMs were gated using CD45-APC Cy7 (Biolgend, San Diego, CA, USA), CD68-APC (Biolgend, San Diego, CA, USA), CD11c-phycoerythrin (CD11b-PE, Biolgend, San Diego, CA, USA), and CD206-FITC (Biolgend, San Diego, CA, USA). KCs were gated using CD45-FITC, F4/80-APC, CD11c-phycoerythrin (CD11b-PE, Biolgend, San Diego, CA, USA), and CD206-FITC (Biolgend, San Diego, CA, USA). Finally, to count the number of ATMs and KCs, we used FACSCanto (BD Bioscience, Franklin Lakes, NJ, USA) after washing and centrifugation and FlowJo software (Tree Star, Inc., Ashland, OR, USA).

#### 4.2.11. Statistical Analysis

All statistical analyses were performed using GraphPad PRISM 5 (GraphPad Software Inc., San Diego, CA, USA). Between-group differences were evaluated with the one-way analysis of variance and Tukey’s post hoc test. All data are expressed as mean ± standard error. Two-tailed *p*-values < 0.05 were set to indicate statistical significance. Statistical significance compared with the NC and HFD groups is presented with number signs (#) and asterisks (*), respectively (# or * for *p* < 0.05; ## or ** for *p* < 0.01; and ### or *** for *p* < 0.001).

## 5. Conclusions

MGF improves metabolic phenotypes, including BW, decreased fat mass and hepatic lipid accumulation, and improved glucose and lipid metabolism. Moreover, MGF modulated macrophages in WAT and liver, and gene expressions related to inflammation and autophagy. Therefore, the study suggests that MGF improves glucose and lipid metabolism by inhibiting inflammation and enhancing autophagy in adipose tissue and the liver. These findings indicate that MGF is potentially applicable in metabolic syndrome, including obesity and type 2 diabetes.

## Figures and Tables

**Figure 1 ijms-23-15329-f001:**
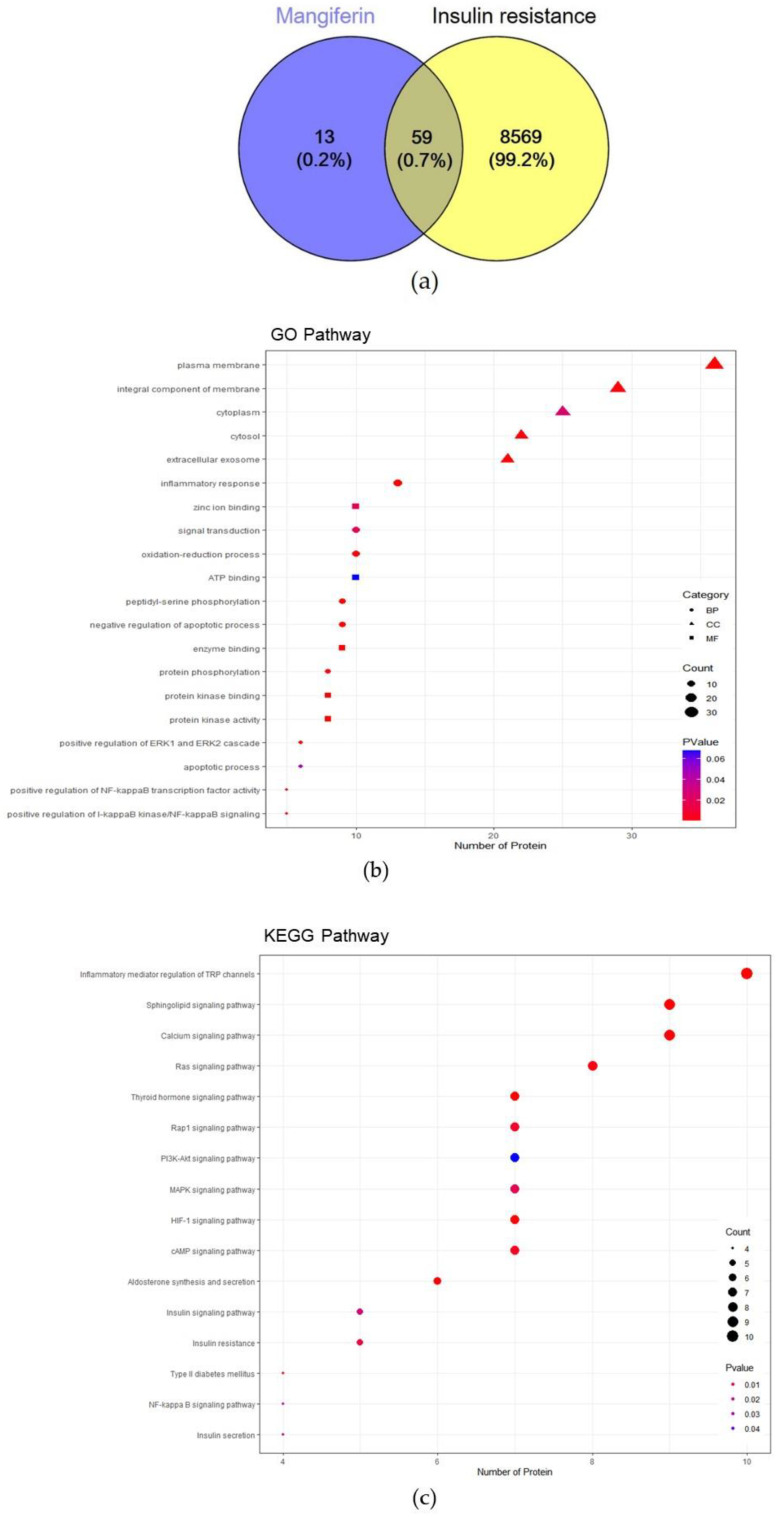
Hub genes of mangiferin (MGF) and insulin resistance and functional enrichment analysis. (**a**) The 59 overlapping genes between MGF and insulin resistance. (**b**) GO enrichment analysis. (**c**) KEGG pathway enrichment analysis. GO, gene ontology; KEGG, Kyoto Encyclopedia of Genes and Genomes.

**Figure 2 ijms-23-15329-f002:**
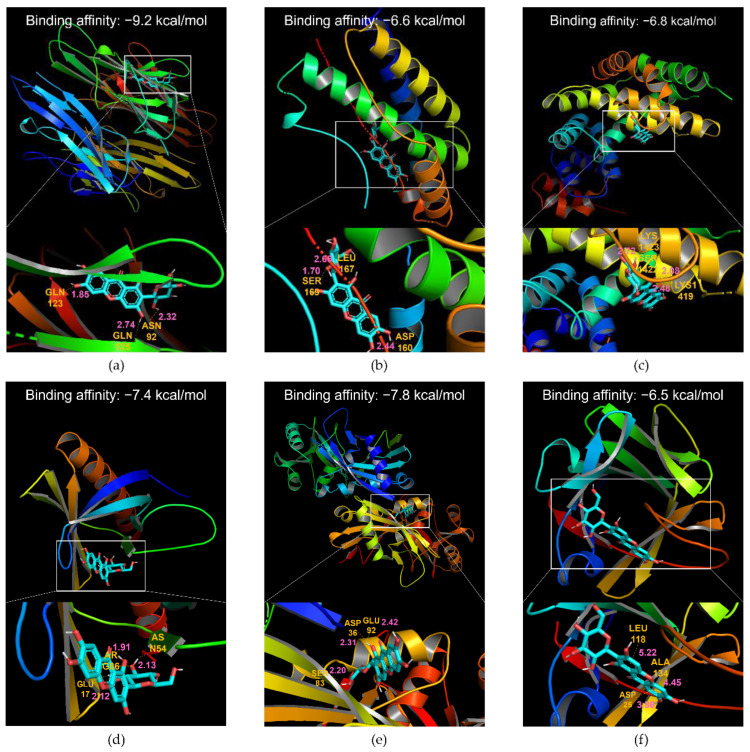
Molecular docking between the MGF and target proteins. (**a**) MGF and TNF-α. (**b**) MGF and IL-6. (**c**) MGF and NF-κB. (**d**) MGF and AKT1. (**e**) MGF and ATG7. (**f**) MGF and FGF21. Binding energies are expressed on the middle top section of each sub-panel. The top three shortest interaction distances are displayed in pink, and the interacting amino acids are shown in yellow. MGF, mangiferin; TNF-α, tumor necrosis factor-α; IL-6, interleukin-6; NF-κB, nuclear factor kappa B; ATG7, autophagy-related gene 7; FGF21, fibroblast growth factor 21.

**Figure 3 ijms-23-15329-f003:**
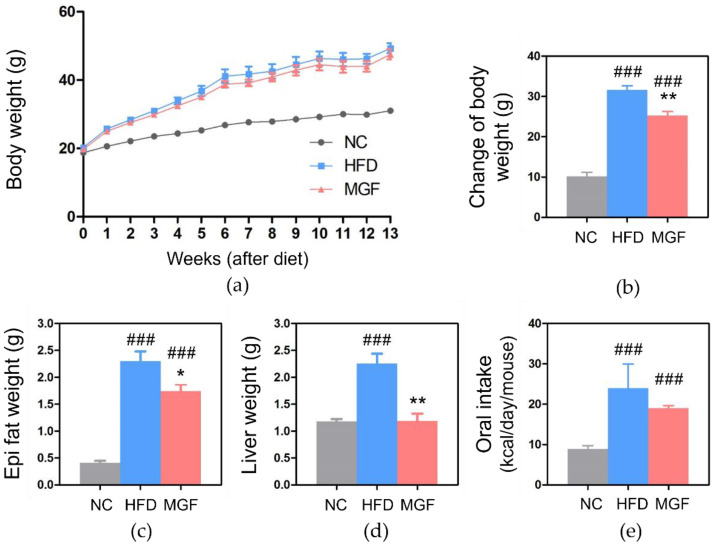
Weight-related indexes of each experimental group (**a**) Timeline of body weight gain. (**b**) Change in body weight. (**c**) Epididymal fat pad weight. (**d**) Liver weight. (**e**) Calorie intake. Data are expressed as mean ± standard error of mean. ### *p* < 0.001 indicates the results of comparison with the NC group, and * *p* < 0.05 and ** *p* < 0.01 indicate the results of comparison with the HFD group. NC, normal chow; HFD, high-fat diet; MGF, HFD + MGF 150 mg/kg/day; MGF, mangiferin.

**Figure 4 ijms-23-15329-f004:**
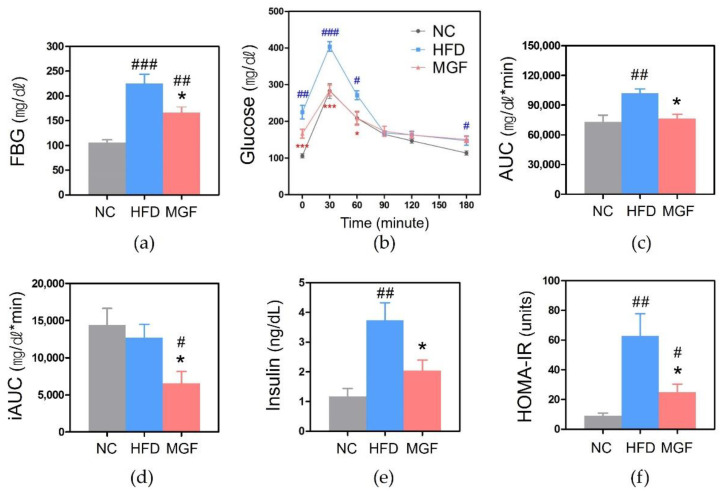
Glucose-metabolism-related outcomes. (**a**) Fasting blood glucose (FBG). (**b**) Oral glucose tolerance test (OGTT). (**c**) AUC of OGTT. (**d**) iAUC of OGTT. (**e**) Serum insulin level. (**f**) HOMA-IR. Data are expressed as mean ± standard error of mean. # *p* < 0.05, ## *p* < 0.01, and ### *p* < 0.001 indicate the results of comparison with the NC group, and * *p* < 0.05 and *** *p* < 0.001 indicate the results of comparison with the HFD group. NC, normal chow; HFD, high-fat diet; MGF, HFD + MGF 150 mg/kg/day; MGF, mangiferin; FBG, fasting blood glucose; AUC, area under the curve; iAUC, incremental area under the curve; OGTT, oral glucose tolerance test; HOMA-IR, homeostatic model assessment for insulin resistance.

**Figure 5 ijms-23-15329-f005:**
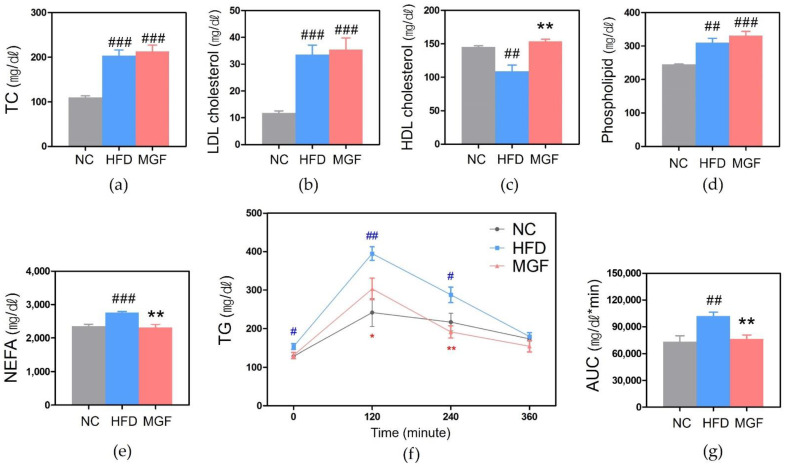
Lipid profiles of each experimental group. (**a**) Total cholesterol (TC). (**b**) LDL cholesterol. (**c**) HDL cholesterol. (**d**) Phospholipid. (**e**) Nonesterified fatty acids (NEFA). (**f**) OFTT. (**g**) AUC of OFTT. Data are expressed as mean ± standard error of mean. # *p* < 0.05, ## *p* < 0.01, and ### *p* < 0.001 indicate the results of comparison with the NC group, and * *p* < 0.05 and ** *p* < 0.01 indicate the results of comparison with the HFD group. NC, normal chow; HFD, high-fat diet; MGF, HFD + MGF 150 mg/kg/day; MGF, mangiferin; TC, total cholesterol; LDL, low-density lipoprotein; HDL, high-density lipoprotein; TG, triglyceride; AUC, area under the curve; OFTT, oral fat tolerance test.

**Figure 6 ijms-23-15329-f006:**
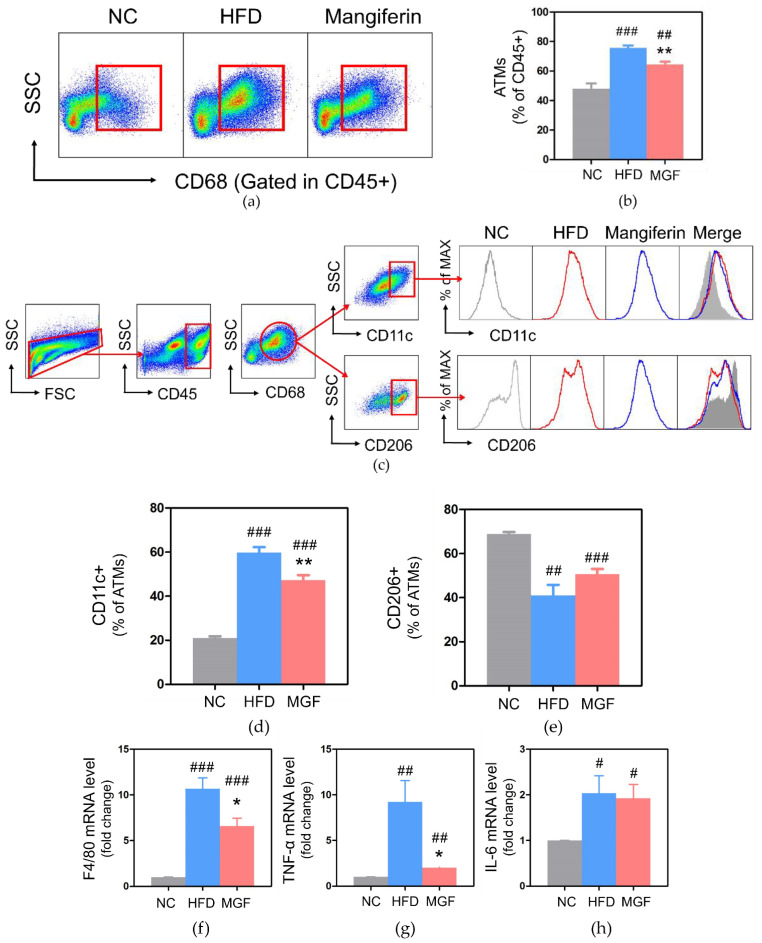
Analysis of ATMs and gene expression of the mRNA level in adipose tissue. (**a**) Flow cytometry of total ATMs. (**b**) Percentage of total ATMs. (**c**) Flow cytometry of CD11c+ and CD206 + ATMs. (**d**) Percentage of CD11c + ATMs. (**e**) Percentage of CD206+ ATMs. Expressions of (**f**) F4/80, (**g**) TNF-α, and (**h**) IL-6. Data are expressed as mean ± standard error of mean. # *p* < 0.05, ## *p* < 0.01, and ### *p* < 0.001 indicate the results of comparison with the NC group, and * *p* < 0.05, ** *p* < 0.01 indicates the results of comparison with the HFD group. NC, normal chow; HFD, high-fat diet; MGF, HFD + MGF 150 mg/kg/day; MGF, mangiferin; ATMs, adipose tissue macrophages; TNF-α, tumor necrosis factor-α; IL-6, interleukin-6.

**Figure 7 ijms-23-15329-f007:**
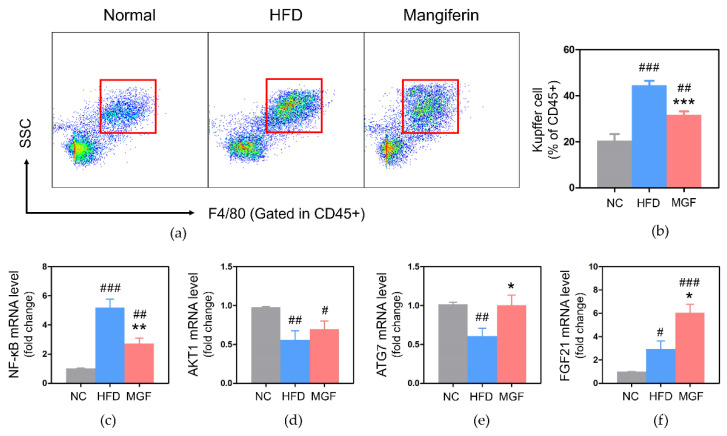
Analysis of KCs and gene expression of the mRNA level in liver tissue. (**a**) Flow cytometry of total KCs. (**b**) Percentage of total KCs. Expressions of (**c**) NF-κB, (**d**) AKT, (**e**) ATG7, and (**f**) FGF21. Data are expressed as mean ± standard error of mean. # *p* < 0.05, ## *p* < 0.01, and ### *p* < 0.001 indicate the results of comparison with the NC group, and * *p* < 0.05 and ** *p* < 0.01, *** *p* < 0.001 indicate the results of comparison with the HFD group. NC, normal chow; HFD, high-fat diet; MGF, HFD + MGF 150 mg/kg/day; MGF, mangiferin; KCs, Kupffer cells; NF-κB, nuclear factor kappa B; ATG7, autophagy-related gene 7; FGF21, fibroblast growth factor 21.

**Figure 8 ijms-23-15329-f008:**
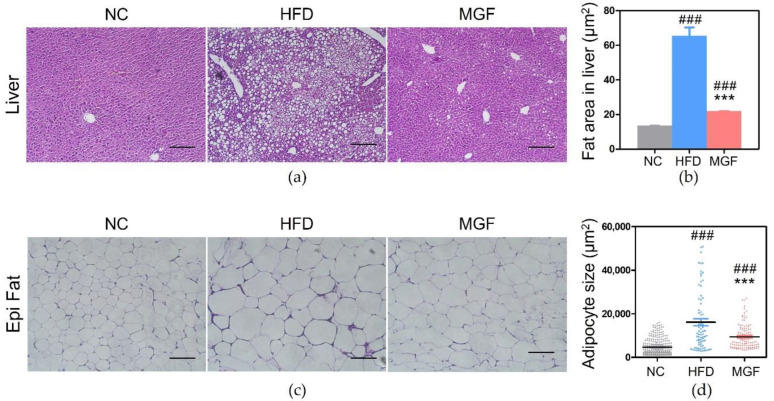
Histological changes in liver and epididymal fat tissues. (**a**) Histological images of the liver. (**b**) Fatty area in the liver. (**c**) Histological images of epididymal fat. (**d**) Adipocyte size. Data are expressed as mean ± standard error of mean. Bar indicates 100 µm. ### *p* < 0.001 indicates the results of comparison with the NC group, and *** *p* < 0.001 indicates the results of comparison with the HFD group. NC, normal chow; HFD, high-fat diet; MGF, HFD + MGF 150 mg/kg/day; MGF, mangiferin.

## Data Availability

Not applicable.

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
