# Peer review of "Mangiferin Ameliorates Obesity-Associated Inflammation and Autophagy in High-Fat-Diet-Fed Mice: In Silico and In Vivo Approaches"

_ijms, 2022, doi:10.3390/ijms232315329_

Round 1
Reviewer 1 Report
Please see the PDF file attached for more detailed comments.
The authors imply that insulin resistance(IR) was improved by their treatment beginning from the title, however, at any point an experiment to evaluate IR in vivo was done. Based on OGTT results, HOMA-IR and gene expression it cannot be said that IR was improved. In addition, insulin signaling was not evaluate at the tissue level by protein activation measurements. The authors should consider the title and affirmations and if IR is their primary outcome another method should be used to evaluate IR.
BW graphs and description does not match and should be checked. This puts in question if all the other results are being appropriately described.
Discussion and language used does not reflect the results, again nothing was done to measure IR. Although improvements in markers of metabolic dysfunction were seen after MGF treatment, the writing and the overall conclusion of each section must be improved to reflect the real meaning of those results considering the method that was not used and its limitations.
Overall, many affirmations are not true, for example, one cannot say that differences in glucose concentration during 0-30 min reflect insulin resistance, because it is simply not true. Basic concepts of glucose homeostasis and insulin resistance are wrongly used throughout the paper and leads to misunderstandings.
In general, writing must be revised to reflects the true results presented.

Author Response
Answers to Reviewer’ Comments
Manuscript No.: ijms-2014135
Authors: Ji-Won Noh et al.
Title: Mangiferin ameliorates obesity-associated inflammation and autophagy in high fat diet-fed mice: In silico and in vivo approaches
Thank you very much for considering our manuscript for publication. Your suggestions were very helpful to us, and we have incorporated those points into our revised manuscript.
The changes made to the manuscript are as follows:
Reviewer 1
Please see the PDF file attached for more detailed comments.
=> We edited manuscript refer to PDF file received. Changed part of manuscript was indicated in red color.
The authors imply that insulin resistance(IR) was improved by their treatment beginning from the title, however, at any point an experiment to evaluate IR in vivo was done. Based on OGTT results, HOMA-IR and gene expression it cannot be said that IR was improved. In addition, insulin signaling was not evaluate at the tissue level by protein activation measurements. The authors should consider the title and affirmations and if IR is their primary outcome another method should be used to evaluate IR.
=> As your comments, insulin resistance is classically defined as the inability of insulin-sensitive organs, such as muscle, fat, and liver, to respond well to insulin and use glucose for energy. This study used the concept of insulin resistance as inflammation because chronic low-grade systemic inflammation due to obesity is a major cause of insulin resistance. Therefore, we mainly investigated the inflammatory cytokines and autophagy mechanisms in liver and adipose tissue and titled it as insulin resistance improved by MGF. However, we totally agree that the basic concept of insulin resistance should be followed as you pointed out, and change the title to inflammation and autophagy instead of insulin resistance.
New title: Mangiferin ameliorates obesity-associated inflammation and autophagy in high-fat diet-fed mice: In silico and in vivo approaches
BW graphs and description does not match and should be checked. This puts in question if all the other results are being appropriately described.
=> As you commented, we checked all the graphs and descriptions. The graphs and descriptions were matched except for BW data. So, we changed the BW graphs.
Figure 3. Weight-related indexes of each experimental group (a) Timeline of body weight gain. (b) Change in body weight. (c) Epididymal fat pad weight. (d) Liver weight. (e) Calorie intake. Data are expressed as mean ± standard error of mean. ###p<0.001 indicates the results of comparison with the NC group, and *p<0.05 and **p<0.01 indicate the results of comparison with the HFD group. NC, normal chow; HFD, high-fat diet; MGF, HFD + MGF 150 mg/kg/day; MGF, mangiferin.
Discussion and language used does not reflect the results, again nothing was done to measure IR. Although improvements in markers of metabolic dysfunction were seen after MGF treatment, the writing and the overall conclusion of each section must be improved to reflect the real meaning of those results considering the method that was not used and its limitations.
Overall, many affirmations are not true, for example, one cannot say that differences in glucose concentration during 0-30 min reflect insulin resistance, because it is simply not true. Basic concepts of glucose homeostasis and insulin resistance are wrongly used throughout the paper and leads to misunderstandings.
=> As you pointed out, we focused on the discussion to reflect our results and to correct misunderstandings.
In general, writing must be revised to reflects the true results presented.
=> As you pointed out, the revised manuscript was proofread again by a professional English editing company (Editage, https://www.editage.com/).
We thank you again for your insightful comments on our paper.
Sincerely yours,
Byung-Cheol Lee, M.D.& Ph.D.
Reviewer 2 Report
Comments to the Author
This study was aimed to investigate the molecular and genetic mechanisms of MGF in insulin resistance, focusing on the inflammatory response and autophagy.
Overall comments
The manuscript requires careful editing for grammar/clarity/spelling.
Here are a few examples:
- Line 30: with an increasing treatment cost in Korea...
- Line 37: responses and insulin signaling disorders via an increased secretion...
- Line 39: Recently, number of studies on autophagy, dysregulation, and metabolic diseases have increased...
- Line 53: However, the role of MGF is mediating autophagy and insulin resistance has not been investigated...
- It is recommended to Improve the quality of figure 1.
Introduction
- It is essential to make a detailed description of the Anemarrhenae Rhizoma plant, including its characteristics, cultivation areas, and level of consumption in other parts of the world.
- Are there studies that evaluate the level of toxicity of the Anemarrhenae Rhizoma plant? If so, how are these effects associated with health?
- The author mentions that mangiferin generates a protective effect against obesity and controls diabetes complications; he must indicate the type of study, the dose used, exposure time, and methodology in general.
- The author mentions that mangiferin generates an anti-inflammatory effect on IRAK1 and MAPK; however, the doses of polyphenol used, the type of study model, exposure time, and possible adverse effects should be indicated.
- The author mentions that mangiferin suppresses mTOR activity in the liver. It should indicate the study model type, exposure time, the dose used, and whether other possible markers were analyzed. Were anthropometric parameters taken into account? Was the intake based on the mangiferin extract or the whole plant?
Results
- Line 116: - What does it mean: "as the first group was taller than the second"...?
- The results referring to body weight can confuse the reader since the numerical values ​​do not coincide with the graph. The information must be organized.
- Line 131: "To weigh how MGF affects glucose metabolism..."
- Line 134: Glucose in the control group is slightly elevated, explain.
- Line 137: The wording of the results is confusing; the numerical values ​​do not match when comparing the MGF vs. HFD groups.
- The initial weight is not reported; therefore, we do not know if the values ​​​​are homogeneous at the beginning.
- The weight results must be shown through a timeline to know the evolution and not only indicate the total gain.
- Results of FBG, Insulin, and HOMA-IR (MGF vs NC) are not mentioned
- Results about NEFA on the graphic are missing
- Triglyceride results at 120 min of MGF versus NC group are not reported. It would be interesting to know the outcome, even after it was insignificant.
- In general, there are higher lipid profile values ​​​​in the MGF versus NC group; however, these are not reported in the description.
- The author mentions that there was no difference in IL-6 (HFD vs MGF); however, it does not report the difference between MGF vs NC.
Discussion
- Check the grammar.
- The author mentions that MGF improved the weight and fat area of ​​the fat pad and liver and the expression of AKT and FGF21 mRNA; however, it should be noted that this effect is compared to a metabolically impaired group on an obesogenic diet (HFD). It is not understood whether the effect is due to resistance against obesity or a direct effect of MGF; therefore, should be tested a supplemented control group.
- The results related to caloric intake are not described; it is interesting because no changes were shown between the HFD and HFD+FGM groups; however, if effects are observed in the parameters of weight and adipose tissue, these effects should be described.
- It would be interesting to analyze mechanisms that regulate intake, such as the neurotransmitters NPY and POMC, due to the results obtained in energy intake.
- The lipid profile results generate certain doubts, As these values ​​are high in the HFD and HFD+FGM groups and are not compared with the control group.
- The results of the lipid profile should be compared with the control group and also use a supplemented control group; this could give a more accurate explanation of the MGF effect.
- The gene expression results are interesting; however, they are inconclusive due to the lack of a control group supplemented with MGF.
Material and methods
- The methodology for measuring caloric intake is not mentioned. How was consumption measured?
- Regarding the body weight measurement, how many people were in charge of the process? It is essential to define it to avoid variability in the results.
- The composition of the diets used, both control and DH, is not mentioned.
- The dietary design of the FGM group is not clearly described.
- The total number of animals used per group is not described.
- The randomization method of the animals by the group is not described.
- The author mentions an administered dose of 150 mg/Kg per day. Are these values ​​relative to the animal's body weight or the daily diet's total weight?
- The dose of 150 mg/Kg used must be justified, and the selection criteria and the history of this dose.
- The macronutrients of the diets used as carbohydrates and lipids are unknown.
Author Response
Answers to Reviewer’ Comments
Manuscript No.: ijms-2014135
Authors: Ji-Won Noh et al.
Title: Mangiferin ameliorates obesity-associated inflammation and autophagy in high fat diet-fed mice: In silico and in vivo approaches
Thank you very much for considering our manuscript for publication. Your suggestions were very helpful to us, and we have incorporated those points into our revised manuscript.
The changes made to the manuscript are as follows:
Overall comments
The manuscript requires careful editing for grammar/clarity/spelling.
Here are a few examples:
- Line 30: with an increasing treatment cost in Korea...
- Line 37: responses and insulin signaling disorders via an increased secretion...
- Line 39: Recently, number of studies on autophagy, dysregulation, and metabolic diseases have increased...
- Line 53: However, the role of MGF is mediating autophagy and insulin resistance has not been investigated...
- It is recommended to Improve the quality of figure 1.
=> We corrected and marked it as red color in manuscript including Line 30, 37, 39, and 53, and changed figure 1 with higher resolution.
Introduction
- It is essential to make a detailed description of the Anemarrhenae Rhizoma plant, including its characteristics, cultivation areas, and level of consumption in other parts of the world.
=> We added Anemarrhenae asphodeloides’ characteristics, cultivation areas, and level of consumption in other parts of the world and marked it as red color.
- Are there studies that evaluate the level of toxicity of the Anemarrhenae Rhizoma plant? If so, how are these effects associated with health?
=> According to previous review study, more attention has been paid to the potential toxic reactions of A. asphodeloides, there were no metabolomic and systematic studies to evaluate the toxicity. Therefore, further research should be conducted to evaluate the potential toxic reactions of A. asphodeloides in animal models and to determine the minimum effective dosage for obtaining
the preclinical safety data.
[Related reference]
Liu,C.; Cong,Z.; Wang,S., et al. A Review of the Botany, Ethnopharmacology, Phytochemistry, Pharmacology, Toxicology and Quality of Anemarrhena Asphodeloides Bunge. Journal of ethnopharmacology 2023 , 302.
- The author mentions that mangiferin generates a protective effect against obesity and controls diabetes complications; he must indicate the type of study, the dose used, exposure time, and methodology in general.
- The author mentions that mangiferin generates an anti-inflammatory effect on IRAK1 and MAPK; however, the doses of polyphenol used, the type of study model, exposure time, and possible adverse effects should be indicated.
- The author mentions that mangiferin suppresses mTOR activity in the liver. It should indicate the study model type, exposure time, the dose used, and whether other possible markers were analyzed. Were anthropometric parameters taken into account? Was the intake based on the mangiferin extract or the whole plant?
=> As you commented, we added type of study, the dose used, exposure time about each study and marked it as red color.
Results
- Line 116: - What does it mean: "as the first group was taller than the second"...?
- The results referring to body weight can confuse the reader since the numerical values ​​do not coincide with the graph. The information must be organized.
- Line 131: "To weigh how MGF affects glucose metabolism..."
- Line 134: Glucose in the control group is slightly elevated, explain.
- Line 137: The wording of the results is confusing; the numerical values ​​do not match when comparing the MGF vs. HFD groups.
=> We corrected and marked it as red color in manuscript including line 116, 131, 134, 137 for grammar, clarity, and spelling.
- The initial weight is not reported; therefore, we do not know if the values ​​​​are homogeneous at the beginning.
- The weight results must be shown through a timeline to know the evolution and not only indicate the total gain.
=> As you pointed out, we added figure of change of body weight after diet.
- Results of FBG, Insulin, and HOMA-IR (MGF vs NC) are not mentioned
- Results about NEFA on the graphic are missing
- Triglyceride results at 120 min of MGF versus NC group are not reported. It would be interesting to know the outcome, even after it was insignificant.
- In general, there are higher lipid profile values ​​​​in the MGF versus NC group; however, these are not reported in the description.
- The author mentions that there was no difference in IL-6 (HFD vs MGF); however, it does not report the difference between MGF vs NC.
=> As you commented, we added figure of results about NEFA. We indicated results of FBG, Insulin, and HOMA-IR, triglyceride results at 120 min, lipid profile, and IL-6 expression (MGF vs. NC).
Discussion
- Check the grammar.
- The author mentions that MGF improved the weight and fat area of ​​the fat pad and liver and the expression of AKT and FGF21 mRNA; however, it should be noted that this effect is compared to a metabolically impaired group on an obesogenic diet (HFD). It is not understood whether the effect is due to resistance against obesity or a direct effect of MGF; therefore, should be tested a supplemented control group.
=> As you commented, we note the comparison between MGF group and HFD group. We agree with you about supplemented control group which could mean a positive control group. The absence of positive control group is a limitation of our study.
- The results related to caloric intake are not described; it is interesting because no changes were shown between the HFD and HFD+FGM groups; however, if effects are observed in the parameters of weight and adipose tissue, these effects should be described.
=> As you pointed out, we realize that we did not describe the weight loss effects along with no food intake difference. So, we add this point into the 5th paragraph in discussion part.
- It would be interesting to analyze mechanisms that regulate intake, such as the neurotransmitters NPY and POMC, due to the results obtained in energy intake.
=> As you commented, Obesity increases inflammation in the hypothalamic arcuate nucleus, which controls appetite and satiety by releasing neuropeptide Y (NPY) and pro-opiomelanocortin (POMC). MGF can reduce neuroinflammation and oxidative damage in the brain [31]. In previous studies, weight loss caused by MGF might have resulted from modulating hypothalamic neuroinflammation and appetite-related neurotransmitters. However, in this study, we focused on macrophage-related inflammatory cytokines and autophagy mechanisms in liver and adipose tissue and did not analyze hypothalamic inflammation. Therefore, further studies are required for confirmation. Also, we add this point into the 5th paragraph in discussion part.
- The lipid profile results generate certain doubts, As these values ​​are high in the HFD and HFD+FGM groups and are not compared with the control group.
=> As you commented, we explain the lipid profile results and additionally compare MGF group with the NC group at 7th paragraph in the discussion part.
- The results of the lipid profile should be compared with the control group and also use a supplemented control group; this could give a more accurate explanation of the MGF effect.
=> We agree with you about supplemented control group which could mean a positive control group. The absence of positive control group is a limitation of our study.
- The gene expression results are interesting; however, they are inconclusive due to the lack of a control group supplemented with MGF.
=> We agree with you about supplemented control group which could mean a positive control group. The absence of positive control group is a limitation of our study.
Material and methods
- The methodology for measuring caloric intake is not mentioned. How was consumption measured?
=> The methodology to measure oral intake in gram was shown 4.2.3. “In oral intake, the quantity consumed by each group was measured every morning by the difference between the day before feed and the following day using an electronic balance.”. The caloric intake was calculated as g*2.91kcal/g for normal chow and g*5.24 kcal/g for high fat diet.
- Regarding the body weight measurement, how many people were in charge of the process? It is essential to define it to avoid variability in the results.
=> Two people were in charge of body weight measurement. One was in charge of positioning mouse and reading the scale, and the other was in charge of writing down the weights.
- The composition of the diets used, both control and DH, is not mentioned.
=> The diet consumed by HFD group and MGF group was D12492 diet (Research Diets, USA) consisting of a fat source of soybean oil and particularly lard. We add the product code of HFD into the manuscript 4.2.1., and supplementary table S1 about a composition of D12492.
- The dietary design of the FGM group is not clearly described.
=> The diet of MGF group was HFD (high-fat-diet) as same as HFD group to induce obesity both in HFD and MGF groups. This explanation was in 4.2.2. “The HFD and MGF groups were administered with HFD for 14 weeks to induce obesity.”
- The total number of animals used per group is not described.
=> The total number of experimental mice was 15, and each group has 5 mice. We describe the total number of mice in 4.2.2. in the manuscript.
- The randomization method of the animals by the group is not described.
=> This in vivo experiment was designed for 8 weeks of treatment of MFG after 6 weeks of high-fat feeding. After one week of adaptation, mice were simultaneously randomized to the three groups without considering any other variable, based on a single sequence of random assignments method.
- The author mentions an administered dose of 150 mg/Kg per day. Are these values ​​relative to the animal's body weight or the daily diet's total weight?
=> The 150 mg/kg was relative to animal’s body weight.
- The dose of 150 mg/Kg used must be justified, and the selection criteria and the history of this dose.
=> According to previous in vivo studies, various doses were used, but the doses under 150 mg/kg did not have consistent effects on weight loss, glucose-lowering and lipid-lowering. So, we selected the dose of 150 mg/kg which was also selected by the other human study.
[Related reference]
- Fomenko, E. V.; Chi, Y. Mangiferin Modulation of Metabolism and Metabolic Syndrome. BioFactors 2016, 42, 492-503.
- The macronutrients of the diets used as carbohydrates and lipids are unknown.
=> We add supplementary table of the composition of ‘D12492 (Research Diets)’.
We thank you again for your insightful comments on our paper.
Sincerely yours,
Byung-Cheol Lee, M.D.& Ph.D.

Round 2
Reviewer 1 Report
Although the authors made improvements compared to the first version, there are still extensive changes to me made.
1. The abstract should be reviewed since it still refers to actions in insulin resistance.
2. Why is the dosage used in this study more than double for the dosage showed in the other cited studies for the compound used?
3. line 36- adipose tissue mass?
4. Line 53- spell out IBD
5. line 95- Can the authors affirm that it is ameliorating insulin resistance or they are related to insulin resistance?
6. line 143-144- incremental AUC should be calculated, considering that there is difference between fasting glucose levels. This will show if the differences seem are dur to differences in glucose tolerance or due to fasting glucose.
7. line 250- what do the authors mean by genetic mechanisms?
8. Line 261- AKT was not increased
9. line 262-How cant he authors affirm that insulin resistance was improved by an in silico approach?
10. line 270-274- sentence needs to be restructured
11. line 279-280- sentence needs to be restructured
12. Line 298- weight loss was not observed, weigh gain was reduced which is very different from weight loss.
13. line 305-306- his doesn't make sense. You are saying that weight loss (which didn't happen) was because of changes in macrophages? macrophages were not manipulated, word choice is important
14. line 308- What do you mean by expansion of WAT in internal organs? It doesn't make sense as well.
15. line 310-312- How does that authors explain that decreased KCs and autophagy lead to decreased fat in the liver and reduced adipocyte size?
16. line 333- incomplete sentence
17. line 337- what is special about it?
18. line 340-It does not promote lipolysis. I actually does not inhibit lipolysis effectively. Word choice is important.
19. line 344- "MGF improved insulin resistance" This is not shown.
20. line 357- "glucose and insulin resistance" - what do you mean by glucose resistance?
21. line 368- there is no evidence of improvements in insulin resistance.
22. line 370- show p value for the trend.
23. line 371-373- weird sentences, change wording
24. line 430- Only in the beginning and final of the trial? or throughout the trial?
25. line 433- "in oral intake" - check wording
26. line 526-529- need to be restructured
Extensive English revisions are required. Even though the authors say that English was revised, there are several typos in the manuscript (some, but not all, were highlighted in the pdf version). In addition, some sentences are badly structured, making it difficult to understand what the authors want to say and causing misunderstandings. Word choice is extremely important when translating your research, please revise and change words appropriately.

Author Response
Answers to Reviewers’ Comments
Manuscript No.: ijms-2014135
Authors: Ji-Won Noh et al.
Title: “Mangiferin ameliorates obesity-associated inflammation and autophagy in high fat diet-fed mice: In silico and in vivo approaches”
Thank you very much for considering our manuscript for publication. Your suggestions were very helpful to us, and we have incorporated those points into our revised manuscript.
The changes made to the manuscript are as follows:
- The abstract should be reviewed since it still refers to actions in insulin resistance.
→ We edited the abstract as you commented.
- Why is the dosage used in this study more than double for the dosage showed in the other cited studies for the compound used?
→ The cited articles in the introduction part explained the anti-inflammatory properties of mangiferin. However, our study goal was to figure out the mangiferin’s anti-obesity effect. According to Wu’s summary of in vivo studies of oral administration of mangiferin, the effects of low doses of mangiferin (10-60 mg/kg BW/day) on abdominal and epididymal fat differed between studies. As a result of comparing mangiferin at three doses of 50, 100, and 150 mg/kg BW with a control group (Niu, 2012), only the group administered with 150 mg/kg BW of mangiferin had significant decrease in abdominal and epididymal fat weight in hyperlipidemic rats. At a genetic level, Guo reported that the significant differences from a control group in liver PPAR-α and SREBP-1c were only seen in the mangiferin group with 150 mg/kg BW dose. Therefore, we chose the oral dose of 150 mg/kg BW/day to reveal the mechanism of the anti-obesity effects of mangiferin.
[Related reference]
- You Wu, Wi Liu, Tao yang, Mei Li, Lingling Qin, Lili Wu, Tonghua Liu. Oral administration of mangiferin ameliorates diabetes in animal models: a meta-analysis and systemic review. Nutrition research 2021, 87, 57-69. https://doi.org/10.1016/j.nutres.2020.12.017
-Yucun Niu, Songtao Li, Rennan Feng, liyan Liu, Ying Li, Changhao Sun. Mangiferin decreases plasma free fatty acids through promoting its catabolism in liver by activation of AMPK. PLoS ONE. 2012, 7(1):e30782. https://doi.org/10.1371/journal.pone.0030782
-Fuchuan Guo, Conghui Huang, Xilu Liao, Yemei Wang, Ying He, Rennan Feng, Ying Li, Changhao Sun. Beneficial effects of mangiferin on hyperlipidemia in high-fat-fed hamsters. Mol Nutr Food Res. 2011, 55, 1809-1818. https://doi.org/10.1002/mnfr.201100392
- line 36- adipose tissue mass?
- Line 53- spell out IBD
→ We corrected line 36 and 53.
- line 95- Can the authors affirm that it is ameliorating insulin resistance or they are related to insulin resistance?
→ As you commented, we could not affirm the amelioration of insulin resistance with an in silico study, but we could infer a link between mangiferin and multiple pathways involved in insulin resistance. So, we edited this point in the manuscript (line 95).
- line 143-144- incremental AUC should be calculated, considering that there is difference between fasting glucose levels. This will show if the differences seem are due to differences in glucose tolerance or due to fasting glucose.
→ As you pointed out, we calculated iAUC and added the matching graph to minimize the variations in fasting plasma glucose or baseline blood glucose.
- line 250- what do the authors mean by genetic mechanisms?
→ The words ‘genetic mechanisms’ could be misunderstood as genetic variations, so we rewrote the sentence in line 250.
- Line 261- AKT was not increased
→ We deleted it.
- 9. line 262-How cant the authors affirm that insulin resistance was improved by an in silico approach?
→ We agree with your comments and correct the sentence.
“Finally, in silico experiments, it was revealed that MGF affected autophagy-related mechanisms such as mTOR signaling pathways among the genes related to insulin resistance.”
- line 270-274- sentence needs to be restructured
- line 279-280- sentence needs to be restructured
→ As you pointed out, we edited line 270-274 and line 279-280.
- Line 298- weight loss was not observed, weigh gain was reduced which is very different from weight loss.
→ We revised it in line 298.
- line 305-306- his doesn't make sense. You are saying that weight loss (which didn't happen) was because of changes in macrophages? macrophages were not manipulated, word choice is important
→ We revised the sentence in line 305-306 to be precise.
“The obesity preventive effects were not based on differences in food intake but on significant immune-modulating benefits, including decreased M1 ATM, inhibition of TNF-α, and enhancement of NF-κB.”
- line 308- What do you mean by expansion of WAT in internal organs? It doesn't make sense as well.
→ As you commented, we corrected line 308.
- line 310-312- How does that authors explain that decreased KCs and autophagy lead to decreased fat in the liver and reduced adipocyte size?
→ We rewrote the sentences in line 309-315 to make clear the effects of MGF on a linkage of KCs, fat in the liver, and adipocyte size and deleted ‘autophagy’ in this sentence.
[Related reference]
- Rinke Stienstra, fredy saudale, Caroline Duval, Shohreh Keshtkar, johanna E. M. Groener, Nico van Rooijen, Bart Staels, Sander Dersten, Michael Muller. Kupffer cells promote hepatic steatosis via interleukin-1β-dependent suppression of peroxisome proliferator-activated receptor α activity. Hepatology. 2010, 51(2), 511-522. https://doi.org/10.1002/hep.23337
- Xuemei Fang, Shanshan zou, Yuanyuan Zhao, Ruina Cui, Wei Zhang, Jiayue Hu, Jiayin Dai. Kupffer cells suppress perfluorononanoic aicd-induced hepatic peroxisome proliferator-activated receptor α expression by releasing cytokines. Archives of Toxicology. 2012, 86, 1515-1525. https://10.1007/s00204-012-0877-4
- line 333- incomplete sentence
→ We completed line 333.
- line 337- what is special about it?
→ We corrected line 337.
- line 340-It does not promote lipolysis. I actually does not inhibit lipolysis effectively. Word choice is important.
→ We changed the word ‘promoting’ to ‘stimulating’.
[Related reference]
- Hui H. Zhnag, Melanie Halbleib, Faiyaz Ahmad, Vincent C. Manganiello, Andrew S. Greenberg. Tumor Necrosis Factor-α Stimulates Lipolysis in Differentiated Human Adipocytes Through Activation of Extracellular Signal-Related Kinase and Elevation of Intracellular cAMP. Diabetes, 2002, 51(10), 2929-2935. https://doi.org/10.2337/diabetes.51.10.2929
- line 344- "MGF improved insulin resistance" This is not shown.
- line 357- "glucose and insulin resistance" - what do you mean by glucose resistance?
- line 368- there is no evidence of improvements in insulin resistance.
→ We edited line 344, 357, and 368.
- line 370- show p value for the trend.
→ We edited it.
- line 371-373- weird sentences, change wording
- line 430- Only in the beginning and final of the trial? or throughout the trial?
- line 433- "in oral intake" - check wording
- line 526-529- need to be restructured
→ As you pointed out, we corrected line 371-373, 430, 433, and 526-529.
Extensive English revisions are required. Even though the authors say that English was revised, there are several typos in the manuscript (some, but not all, were highlighted in the pdf version). In addition, some sentences are badly structured, making it difficult to understand what the authors want to say and causing misunderstandings. Word choice is extremely important when translating your research, please revise and change words appropriately.
→This manuscript has been edited by a professional English language editing company (editage: https://www.editage.co.kr/).
We thank you again for your insightful comments on our paper.
Sincerely yours,
Byung-Cheol Lee, M.D.& Ph.D.

Reviewer 2 Report
- Check the grammar.
Author Response
Answers to Reviewers’ Comments
Manuscript No.: ijms-2014135
Authors: Ji-Won Noh et al.
Title: Mangiferin ameliorates obesity-associated inflammation and autophagy in high fat diet-fed mice: In silico and in vivo approaches
Thank you very much for considering our manuscript for publication. Your suggestions were very helpful to us, and we have incorporated those points into our revised manuscript. This manuscript has also been edited by a professional English language editing company (editage: https://www.editage.co.kr/).
We thank you again for your insightful comments on our paper.
Sincerely yours,
Byung-Cheol Lee, M.D.& Ph.D.
Round 3
Reviewer 1 Report
Please see below
Line 18 – fix: organs weight
Line 33- sentence restructure “obesity-induced insulin resistance”
Line 36/37- -sentence restructure “impair insulin signaling”
Line 45- remove “reported”
Line 46- remove “reportedly”
Line 49 remove “demonstrably”
Line 59 – “ between autophagy and insulin resistance” – should be changedà this is not the purpose of the study as the authors already fixed in the abstract.
Line 115 – the subheading is not appropriate- change insulin resistance to glucose and lipid metabolism
Line 117 – fix : group food intake (no ‘)- check whole manuscript for this
Line 283- fix : and renal fibrosis inhibition via PTEN/PIEK/AKT pathway have been reported.
Line 285: fix: MGF effect ( no ‘)
Line 285: fix: possible causes
Line 285/285: change “including fat accumulation in the liver and fat “ to “ including increased fat mass and and hepatic lipid accumulation”
Line 302: remove “in liver” ( haptic glucose output already sys it is in the liver)
Line 310 : “their sensitivity to insulin resistance” – weird sentence, please restructure
Line 322- fix: decreased adipocyte size
Line524-526- replace “MGF improves metabolic phenotypes, including BW, glucose and lipid metabolism, 524 and fat accumulation of liver and adipose tissues. Moreover, it modulates macrophages 525 in adipose tissue and liver, and gene expressions related to inflammation and autophagy” for
“MGF improves metabolic phenotypes, including BW, decreased fat mass and haptic lipid accumulation, and improved glucose and lipid metabolism. Moreover, MGF modulated macrophages in WAT and liver, and gene expression related to inflammation and autophagy”
Author Response
Answers to Reviewers’ Comments
Manuscript No.: ijms-2014135
Authors: Ji-Won Noh et al.
Title: “Mangiferin ameliorates obesity-associated inflammation and autophagy in high fat diet-fed mice: In silico and in vivo approaches”
Thank you very much for considering our manuscript for publication. Your suggestions were very helpful to us, and we have incorporated those points into our revised manuscript.
The changes made to the manuscript are as follows:
[3rd Round Reviewer Comments]
Please see below
Line 18 – fix: organs weight
Line 33- sentence restructure “obesity-induced insulin resistance”
Line 36/37- -sentence restructure “impair insulin signaling”
Line 45- remove “reported”
Line 46- remove “reportedly”
Line 49 remove “demonstrably”
→ We appreciated your thoughtful and detailed comments. We fixed the sentences in the manuscript.
Line 59 – “ between autophagy and insulin resistance” – should be changedà this is not the purpose of the study as the authors already fixed in the abstract.
→ We edit the sentence as follows: “However, few studies investigated the role of MGF treating obesity in aspect of inflammation and autophagy.”
Line 115 – the subheading is not appropriate- change insulin resistance to glucose and lipid metabolism
Line 117 – fix : group food intake (no ‘)- check whole manuscript for this
Line 283- fix : and renal fibrosis inhibition via PTEN/PIEK/AKT pathway have been reported.
Line 285: fix: MGF effect ( no ‘)
Line 285: fix: possible causes
Line 285/285: change “including fat accumulation in the liver and fat “ to “ including increased fat mass and and hepatic lipid accumulation”
Line 302: remove “in liver” ( haptic glucose output already sys it is in the liver)
Line 310 : “their sensitivity to insulin resistance” – weird sentence, please restructure
Line 322- fix: decreased adipocyte size
Line524-526- replace “MGF improves metabolic phenotypes, including BW, glucose and lipid metabolism, 524 and fat accumulation of liver and adipose tissues. Moreover, it modulates macrophages 525 in adipose tissue and liver, and gene expressions related to inflammation and autophagy” for
“MGF improves metabolic phenotypes, including BW, decreased fat mass and haptic lipid accumulation, and improved glucose and lipid metabolism. Moreover, MGF modulated macrophages in WAT and liver, and gene expression related to inflammation and autophagy”
→ We fixed all points as you commented.
We thank you again for your insightful comments on our paper.
Sincerely yours,
Byung-Cheol Lee, M.D.& Ph.D.
